# Effect of the Nanorough Surface of TiO_2_ Thin Films on the Compatibility with Endothelial Cells

**DOI:** 10.3390/ijms24076699

**Published:** 2023-04-03

**Authors:** Irina Yu. Zhuravleva, Maria A. Surovtseva, Andrey A. Vaver, Evgeny A. Suprun, Irina I. Kim, Natalia A. Bondarenko, Oleg S. Kuzmin, Alexander P. Mayorov, Olga V. Poveshchenko

**Affiliations:** 1E. Meshalkin National Medical Research Center, RF Ministry of Health, 15 Rechkunovskaya St., 630055 Novosibirsk, Russia; 2Research Institute of Clinical and Experimental Lymphology, Branch of the Federal Research Center Institute of Cytology and Genetics SB RAS, 2 Timakova St., 630060 Novosibirsk, Russia; 3Boreskov Institute of Catalysis SB RAS, Lavrentiev Ave. 5, 630090 Novosibirsk, Russia; 4Institute of Strength Physics and Materials Science, Siberian Branch Russian Academy of Sciences, 2/4, pr. Akademicheskii, 634055 Tomsk, Russia; 5VIP Technologies Ltd., 634055 Tomsk, Russia; 6Institute of Laser Physics of Siberian Branch, Russian Academy of Sciences, 15B Lavrentiev Av., 630090 Novosibirsk, Russia

**Keywords:** titanium oxide, titanium oxynitride, thin films, nanoroughness, cytocompatibility, reactive magnetron sputtering

## Abstract

The cytocompatibility of titanium oxides (TiO_2_) and oxynitrides (N-TiO_2_, TiO_x_N_y_) thin films depends heavily on the surface topography. Considering that the initial relief of the substrate and the coating are summed up in the final topography of the surface, it can be expected that the same sputtering modes result in different surface topography if the substrate differs. Here, we investigated the problem by examining 16 groups of samples differing in surface topography; 8 of them were hand-abraded and 8 were machine-polished. Magnetron sputtering was performed in a reaction gas medium with various N_2_:O_2_ ratios and bias voltages. Abraded and polished uncoated samples served as controls. The surfaces were studied using atomic force microscopy (AFM). The cytocompatibility of coatings was evaluated in terms of cytotoxicity, adhesion, viability, and NO production. It has been shown that the cytocompatibility of thin films largely depends on the surface nanostructure. Both excessively low and excessively high density of peaks, high and low kurtosis of height distribution (S_ku_), and low rates of mean summit curvature (S_sc_) have a negative effect. Optimal cytocompatibility was demonstrated by abraded surface with a TiO_x_N_y_ thin film sputtered at N_2_:O_2_ = 1:1 and U_b_ = 0 V. The nanopeaks of this surface had a maximum height, a density of about 0.5 per 1 µm^2^, S_ku_ from 4 to 5, and an S_sc_ greater than 0.6. We believe that the excessive sharpness of surface nanostructures formed during magnetron sputtering of TiO_2_ and N-TiO_2_ films, especially at a high density of these structures, prevents both adhesion of endothelial cells, and their further proliferation and functioning. This effect is apparently due to damage to the cell membrane. At low height, kurtosis, and peak density, the main factor affecting the cell/surface interface is inefficient cell adhesion.

## 1. Introduction

Interactions between the cell and the surface largely depend on the features of the surface at the micro- and nano-levels. Quite a lot of information about such features has been accumulated. Surfaces can be characterized by the location, height, and shape of surface nanostructures, and the distance between them [1,2,3]. It is known that submicron density and height of nanostructures with the smoothed shape of the top are most favorable for cells’ functioning [4,5]. It should be noted here that the preferable parameters of surfaces for cells’ functioning largely depend on the cell type contacting with the surface [1,6]. For example, endothelial cells and fibroblasts behave differently in the same interface [7]. Optimal surface characteristics are well understood for bone and immune system cells (e.g., [8,9]). For other types of cells, in particular endotheliocytes, there is no certainty. Taking into account the vast possibilities of manufacturing and modifying various surface topographies [1], it is not possible to collect such information for all possible surfaces. At the same time, these data would provide good opportunities for directed manipulation of the surface structure.

The surface material also performs an important role for the cells’ functioning on the surface [1,10]. A significant part of the biocompatible materials was discovered empirically many years ago, and only modern instrumentation makes it possible to study the mechanisms of this biocompatibility. These materials include TiO_2_, which has been successfully used in various fields of medicine for more than 30 years [11].

TiO_2_ and N-doped TiO_2_ coatings are widely used for surface modification of coronary stents. It has been proven that their hemo- and bio-compatibilities determine good long-term clinical results, which are better than those with limus-eluting stents [12]. While the clinical aspects of these coatings are well understood, there is no biomedical interpretation of their operation, and only recently several papers were published on the hemo- and cyto-compatibility of TiO_2_ thin films [13,14].

The present study is devoted to the effect of nanostructure on the cells’ compatibility for TiO_2_ and N-doped TiO_2_ thin films, prepared by magnetron sputtered on a nitinol substrate. This method of surface fabrication belongs to stochastic patterning techniques [1] and, unlike 3D printing or lithography, cannot provide a preassigned surface structure. We hypothesize that the final structure and properties of the surface are affected by both the initial topography of the surface (valley depths and peak heights) and the sputtering method. Therefore, the initial surface for deposition should be selected in such a way that the structure and composition of the prepared thin film provide a final surface with optimal cytocompatibility. We have previously shown that varying the sputtering modes makes it possible to obtain surfaces of various structures and compositions, both elemental and crystalline (rutile or anatase) [15,16]. Regardless of these varieties, all coatings improved the adhesion and proliferation of endothelial cells compared to control bare nitinol. We also noted that the magnitude of the effect significantly depends on the deposition regime. This effect might be associated with both the TiO_2_ crystalline phase and the structural topography of the surface [17,18]. It is reasonable to assume that the topography of two thin films sputtered on substrates with different topography in the same mode should be different.

In this work, we compare the endothelial cells’ compatibility with TiO_2_ and TiO_x_N_y_ thin films sputtered in various modes on nitinol substrates, which differs in the surface roughness.

## 2. Results

### 2.1. Sample Surface Roughness and Topography

The samples were coated by reactive magnetron sputtering of titanium in a reaction gas medium with various N_2_:O_2_ ratios (Table 1) using a TION-2M setup developed by VIP technologies Ltd. (Tomsk, Russia) [19]. The fixed parameters of sputtering were magnetron discharge power 3 kW, dual system operating frequency 40 kHz, plasma-forming gas (Argon) pressure 0.065 Pa, and an exposure time of 45 min.

The results of the AFM characteristics of studied surfaces are presented in Table 2 and Table 3, and Appendix A. Comparative analysis demonstrated that the root-mean squared roughness was lower for polished samples than that for abraded counterparts (*p* < 0.001).

It was also found that machine-polished C2 sample had the lowest root-mean square roughness (41.95 nm) among all the samples (*p* < 0.05). RMSRs of the coated surfaces were always higher than that of the control polished samples (*p* < 0.05). In the group of the abraded samples, the same trend was revealed, but the differences between the control and samples No. 1, 2, 7, and 8 were not significant (*p* > 0.05). The highest RMSR among all the samples (127.42 nm) was obtained for abraded sample 4 sputtered at N_2_:O_2_ =1:1 and U_b_ = 0 V, but its polished counterpart (sample 12) featured the lowest RMSR among all the coated samples (51.84 nm).

The maximum height of the surface (height gradient, S_z_) and the highest maximum peak height (S_p_) of the sample surfaces demonstrated the same patterns. They were significantly lower for machine-polished C2 as compared to abraded C1 (*p* < 0.05). The highest S_z_ and S_p_ were observed in sample 4, and they significantly differed from all other samples except sample 5 (*p* < 0.05). The smallest S_z_ and S_p_ among all coated samples were found in sample 9 (non-significant differences (*p* > 0.05) with samples 10, 12, and 15).

Kurtosis of height distribution (S_ku_) defines how spiky the distribution of height is for most of the peaks in the measured area (Figure 1). If kurtosis is equal to 3, then it means the peaks are normally distributed and have no sharp and jagged parts. If it is more than 3, then the peaks in this area are mostly pointed; and if it is less than 3, then the peaks are smoothed and height distribution is skewed above the mean plane [20]. The Kurtosis of all samples (except sample 1) exceeded 3, that means the peaks on these surfaces are pointed. Sku of C1 was significantly higher than that of C2 (*p* < 0.034). Interestingly, the Sku of most coated surfaces was the same regardless of preprocessing and ranged from 3.22 to 4.44. Among all the samples, the highest S_ku_ were found for the samples C1 (8.01) and No.14 (6.22).

Mean summit curvature (S_sc_) defines how spiky are the summit’s points of contact with other surfaces [20]. “Summit” is defined in the same way as for the density of summits of the surface—it is any point that is higher than the 8 nearest peaks. The higher the mean curvature value is, the more rounded the shape of the highest summit area. Therefore, the lower the average curvature is, the sharper the top. S_sc_ does not differ significantly between abraded and polished samples (*p* > 0.05). When comparing control samples, it was found that the S_sc_ of abraded C1 was significantly higher than that of the polished C2 (*p* < 0.003). The highest S_sc_ among all coated samples was detected in sample 4 (0.63 μm^−1^), but it did not significantly differ from samples 3 and 15 (*p* > 0.05). The lowest Ssc was obtained for sample 10 (0.04 μm^−1^).

The density of summits of the surface (S_ds_) defines the number of summits per surface area unit. We obtained the following data: machine polished C2 had a significantly lower S_ds_ than abraded C1 (*p* = 0.001). Overall, S_ds_ values did not significantly differ between most abraded and polished samples (*p* > 0.05). Samples 14 and 16 were significantly different (*p* < 0.05) from all other samples with the highest density, and sample 10 was the least dense (*p* < 0.05) among all samples.

Many traces from abrading or polishing were observed on the SEM images of the sample surfaces, (Figure 2). Samples 1–8 visually have significantly more rough surfaces than samples 9–16, which is in complete agreement with the AFM data. Most of the samples had multiple grains on its surfaces, most likely TiO_2_ and TiO_x_N_y_ crystals. The abraded samples No.7 and No.8 demonstrated honeycomb-like structures that completely change the surface topography. The surface of these samples looks more amorphous, and samples Nos. 15 and 16, also deposited at the ratio N_2_:O_2_ = 3:1.

### 2.2. Results of the Cytocompatibility Tests

#### 2.2.1. Indirect Cytotoxicity

The viability of endothelial cells after 24 h incubation with abraded sample extracts was 82–113% (Table 4, Appendix A). In the case of polished samples, the viability was 100–112% (for coated samples), and 115% for bare NiTi.

By the third day, an increase in cell viability was revealed for samples 6–8 and abraded bare NiTi. For polished samples Nos. 9, 14, and bare NiTi, on the contrary, a downward trend was observed. Cell viability with abraded sample extracts 6 and 7 (111% and 123%, respectively) was higher compared to their polished counterparts (85% and 99%, respectively) (*p* < 0.05) (Appendix A). However, if we evaluate the results as a whole, all the studied samples were not toxic to endothelial cells as the data complied with the requirements of the ISO 10993-5 standard (80% living cells) [21].

#### 2.2.2. Adhesion of Endothelial Cells on Abraded and Polished TiO_2_ and TiO_x_N_y_ Samples

Endothelial cells were adhered to all abraded and polished surfaces of the TiO_2_ and TiO_x_N_y_ samples; they had a polygonal shape with reticulate actin microfilaments and formed filopodia (Figure 3). In samples 4, 7, 12, and 13, a continuous network of actin filaments was visualized due to the large number of adherent cells. On the C2 polished surfaces, we observed some cells with actin fibers network and stress fibrils.

The number of adhered endothelial cells after 3 days was higher on the polished surfaces than that on the abraded ones (excluding N-TiO_2_-coated samples in pairs 3–11 and 4–12) (Table 5, Appendix A). After 6 days, the number of adhered cells increased or at least remained the same, except for samples Nos. 3 and 5 (*p* > 0.05), and for controls (*p* < 0.05).

Analyzing the results, we evaluated the adhesion on the third day relative to the control surfaces (100% was the number of adhered cells on the C1 for abraded samples and on the C2 for polished ones). Adhesion was low for samples Nos. 1, 6, 11, and 12.

#### 2.2.3. Proliferation of Endothelial Cells on Abraded and Polished TiO_2_ and TiO_x_N_y_ Samples

The EA.hy926 cells did not proliferate on the bare NiTi controls. Poor proliferation was also observed on abraded samples Nos. 1, 3, and 5. There is a tendency for PI to be higher on polished surfaces than on abraded ones (Table 6, Appendix A). However, the proliferation index for polished samples 10, 14, and 16 was lower than for abraded counterparts. We believe that a significant PI cannot be less than 1.5; therefore, proliferation on the samples Nos. 1, 3, 5, 7, 10, and 14–16 was regarded as low.

#### 2.2.4. Viability of Endothelial Cells TiO_2_ and TiO_x_N_y_ Samples

The viability of the endothelial cells on the abraded surfaces after 3 days was below 80%, except for sample 1 (84%) (Figure 4 and Appendix A, and Table 7). By the sixth day, the alive cell percentage had increased for all coated samples to 76–94%, but decreased for bare NiTi C1 (58%).

For polished surfaces Nos.10–12, 15, and 16, cell viability after 3 days was higher compared to abraded counterparts (*p* < 0.05) (Table 7 and Appendix A). However, the polished samples showed an upward trend by the seventh day, while the polished samples showed a downward trend (with the exception of No.11 and C2).

#### 2.2.5. NO Production of Endothelial Cells on Abraded and Polished TiO_2_ and TiO_x_N_y_ Samples

For abraded samples (No.1–8), the NO production level was comparable to that observed for the polished samples (No.9–16) (Table 8, Appendix A). The highest NO production was observed for samples 8, 9, 15, and 16 (48.5–56.2; μM/mL) (Table 8 and Appendix A). The lowest NO production level was observed for samples 5, 6, 13, and 14. For samples 1–4 and 9–12, the NO production level was comparable to the control.

### 2.3. Cytocompatibility by DTA Results

A decision tree with a hierarchical sequential assessment of the significance of each biological test was used to select TiO_2_ and TiO_x_N_y_ coatings with optimal cytocompatibility (Figure 5). For integral assessing the biocompatibility of the TiO_2_ and TiO_x_N_y_ coatings, the decision tree-like graph visualization, developed earlier [16], was chosen. We used the boundary values of the tests as the branch node (Table 9), and the overall level of thin film cytocompatibility as the predicted outcome.

The level of adhesion on abraded and polished surfaces of coated NiTi samples was taken as the main criterion of biological significance. Subsequent criteria: proliferation index, viability, and NO production by endothelial cells on TiO_2_ and TiO_x_N_y_ samples were taken equally. Each tree node represented a parameter range of the selected criterion. Each “leaf” (box) indicated the sample number and its class membership («yes» or «no»). Leaf color reflected the appropriate parameter levels: red was unacceptable level, yellow was moderate, and green was optimal.

The final decision was substantiated by the following interpretation of the considered set of characteristics:-Optimal cytocompatibility has a surface with no red leaf, and at least 2 green leaves;-A surface was considered acceptable with its 3 green or yellow leaves;-The cytocompatibility was classified uncertain when 2 leaves are red;-The surface was determined unacceptable when 3 leaves are red.

Since indirect cytotoxicity was low for all samples, we did not use it as a criterion when constructing the decision tree. Adhesion was used as the main criterion for evaluating the effect of nanoroughness of the surface on the compatibility with endothelial cells (Figure 5). The use of this criterion allowed us to grade the studied samples into three groups. The group with low adhesion included samples 1, 6, 9–12 and 14; with medium adhesion: 2, 4, 8, 15, and 16; and with high adhesion: 3, 5, 7, 13.

Samples with low cell adhesion constituted the “unacceptable” coating group, since No.1, 10, and 14 had a low proliferation index (<1.5), samples 9, 10, 12, and 14 had low viability, and samples 1, 6, 9, 10, and 14 had low NO production. Sample 12 with low adhesion and viability, but high proliferative index and NO production constituted the “uncertain” coating group. Sample 11 with low adhesion, but high proliferation, viability >85%, and high NO was included in the “acceptable” coating group. The “uncertain” coating group consisted mainly of samples with moderate adhesion (2, 15, and 16), and samples with high adhesion (3, 5, 7, and 13).

In the group of samples with high adhesion (3, 5, 7, and 13), only sample 13 had a high proliferation index, sample 5 had a high viability, and samples 3 and 7 had a high NO production. Therefore, samples of this group constituted the “acceptable” coating group. When constructing the decision tree, sample 4 was included in the group of “optimal” coating according to the criteria for constructing the decision tree (Table 9). Despite moderate adhesion, sample 4 showed a high level of proliferation, high viability and NO production.

## 3. Discussion

Our results confirmed the hypothesis that the cytocompatibility of thin films depends, among other things, on the pretreatment of the nitinol substrate. The surface of machine-polished substrates was significantly smoother than hand abraded ones (Table 1 and Appendix A). Accordingly, the relief of thin films obtained by magnetron sputtering is also different. This is most fully characterized by the height gradient (the distance between maximal valley depth and peak height). For polished surfaces in general, this gradient is 1.5–2 times lower than for abraded ones.

However, this effect is not positive for cytocompatibility. The best result was demonstrated by the initially abraded surface of sample No. 4, sputtered at N_2_:O_2_ = 1:1, and U_b_ = 0 V. Its polished counterpart (sample No. 12) fell into the group of “uncertain”, since it showed poor adhesion and survival. Only one polished sample No. 11 turned out to be in the “acceptable” group, which was somewhat unexpected, since it was sputtered at N_2_:O_2_ = 1:1 and U_b_ = −100 V. Previously, we showed that the crystal structure of thin films sputtered at −100 V are mainly represented by anatase [15,16]. It is well known that anatase is significantly higher cytotoxic than rutile [17,18]. We also found this in the direct cytotoxicity test [16]. Therefore, we considered it reasonable that the surfaces formed at −100 V belong to the group of “unacceptable” (samples 1 and 9) or “uncertain” (Nos. 3, 5, 7, 13, and 15). However, we searched for the reason for the unsatisfactory cytocompatibility of thin films Nos. 2, 6, 10, 12, 14, and 16 in the topographic features of the surfaces. The surface topography cannot be predetermined, since the magnetron sputtering method is semi-random, which leads to stochastic patterning [1]. There are other methods for forming TiO_2_ thin films: thermovacuum, electrochemical, thermochemical, sol-gel method, etc. However, most of them require high temperatures or other aggressive factors, which is highly undesirable for cardiovascular nitinol stents. They are shaped and acquire certain phase transition temperatures also under the high temperature. The additional high temperature exposure can distort complex shape and/or martensite-austenite features of the stents. The advantage of low-temperature magnetron sputtering is the sufficient uniformity of the thin film over the entire surface of a complex-shape object. Moreover, the magnetron sputtered thin film is very strongly bonded to the substrate [22]. Although we cannot design the surface topography to achieve optimal cytocompatibility, we can characterize the surface numerically using SEM or AFM [10,11,23], and determine the parameters that adversely affect its cytocompatibility. Of greatest interest is the characterization of TiO_2_ and TiO_x_N_y_ surfaces in terms of height, sharpness, and density of high-aspect-ratio nanostructures [1]. Most of these structures in the obtained thin films are cone-shaped formations with a height from 200 nm to 1400 nm (Appendix A); they differ in mean summit curvatures, kurtoses, and in the densities of summits, which varies from 0.08 to 0.93 per 1 µm^2^ (Table 2 and Table 3).

The surface No. 4 is the most cytocompatible. This surface has 0.5 peaks/1 μm^2^, peak heights from 600 nm to 1400 nm, and smoothed tops (Figure 1). This patterned surface ensures good adhesion of endothelial cells. Most likely, at the highest height and curvature of the peaks, cells engulf nanopillars with medium adhesion depth (“medium state,” [24]) without membrane penetration [25]. The surfaces of samples Nos. 2 and 8 are similar in shape of structures and provide good adhesion, being significantly lower in height.

We observed low adhesion on samples 6, 10, 12, and 14. Surfaces 6 and 10 were characterized by a very low peak density (less than 0.2 per 1 µm^2^), while surfaces Nos. 14 and 16, on the contrary, had very high density (0.74 and 0.93 per 1 µm^2^). Thus, our data confirm the previously obtained results [1,4,5] that both excessively low (less than 0.25) and excessively high (more than 0.7) peak densities prevent cells’ adhering to the surface. Moreover, surface No.10 with the lowest peak density and surfaces Nos. 14 and 16 with the highest density showed poor proliferation and survival. NO production as a cell functioning characteristics was high in sample 16, but we previously discussed that this may be due to NO releasing from a TiO_x_N_y_ thin films sputtered at N_2_: O_2_ = 3:1. The surface of sample 6 was very heterogeneous and, along with areas of low density, contained areas with a normal density of about 0.3–0.37 per 1 µm^2^ (Table 3). Apparently, cells attach well and proliferate in the areas with normal density, while in more rarefied areas, all cellular functions are inhibited. This is confirmed by the image in Figure 2.

The shape-induced reasons for the unsatisfactory cytocompatibility of samples 2 and 12 are less obvious. For sample 12, possible explanations (see Table 1) could be related to the high heterogeneity of peak heights (IQR of S_p_ was from 362 nm to 687 nm) at one of the lowest surface gradients S_z_ (~505), while the peaks had a sharp top (S_sc_ 0,16). We assume that the primary adhesion of endothelial cells on such a surface is insufficient both due to the difficulty of contact with an excessive smooth relief and the high risk of nanopillar penetration of the first monolayer cells [22]. Further stimulation of cell proliferation (Table 6) may be associated with the paracrine effect of growth factors from non-adherent and dead cells. This causes the formation of cell layers on the surface of the sample [5]. Such cell layers are well visualized when stained with phalloidin (Figure 2) and provide good NO production (Table 8). Poor viability and NO production of sample 2 cannot be explained by the surface topography, the numerical characteristics of which are quite satisfactory. The most likely reason is that this thin film consists of TiO_2_, not of TiO_x_N_y_, unlike other coatings. It is well known that it is TiO_x_N_y_, not TiO_2_ coatings that have high hemo- and bio-compatibility [12,23].

## 4. Materials and Methods

Nitinol (NiTi, Ni 55.8%, Confluent Medical Technologies, USA) flat samples (8 mm × 8 mm × 0.5 mm) were used as substrates.

### 4.1. Sample Preprocessing

#### 4.1.1. Abrading

The samples of groups 1–8 (Table 1) were sequentially abraded with a paper-backed SiC abrasive with crystal sizes P600, P1000, and P1500 (ISO 6344), then cleaned with alcohol in an ultrasonic bath. Uncoated samples abraded by this method were the first control group (C1).

#### 4.1.2. Polishing

The samples of groups 9–16 (Table 1) were machinery polished using a 4PD-200 grinding and polishing machine. A flat brass washer of 200 mm diameter was poured with polishing resin SP-3 with a layer of 3 mm, divided into segments 40 × 40 mm in size. This washer was screwed onto one of the spindles of the machine. The samples were glued onto a 150 mm-diameter brass disk with an on CH-2 wax adhesive resin. Grinding was carried out sequentially with M28 and M14 micropowders from electrocorundum until a smooth matte surface without scratches and gaps was obtained. For polishing, diamond micropowders of grades AM 7/5 and AM 2/1 were alternately used until a smooth mirror surface was obtained. The nitinol plates were unstuck by heating a brass disk on an electric stove, and the adhesive resin was washed off in organic solvents—benzene and acetone.

Uncoated samples polished by this method were in the second control group (C2).

### 4.2. Atomic Force Microscopy (AFM)

An Atomic Force Microscope NTEGRA II (NT-MDT Spectrum Instruments, Moscow, Russia) was used to visualize surfaces (n = 5 in each group) and measure their roughness parameters. Each sample was analyzed under semi-contact mode using a silicone ultrasharp cantilever HA_FM A (NT-MDT Spectrum Instruments, Moscow, Russia). Twenty AFM fields (100 μm × 100 μm, 40 μm × 40 μm, 20 μm × 20 μm, and 5 μm × 5 μm, respectively) were analyzed for each sample and the scan rate was chosen to be 0.7 Hz. Image analysis software (Nova-Px) was used to generate 3D topography images and to compare the root-mean-square (RMSR), kurtosis of height distribution (S_ku_), maximum peak height (S_p_), maximum valley depth (S_v_), density of summits of the surface (S_ds_), and mean summit curvature (S_sc_) of the samples obtained by the software. RMSRs, S_p_, S_v_, and S_ds_ parameters were chosen to be measured on 100 μm × 100 μm fields; S_ku_ and S_sc_ were measured on 40 μm × 40 μm fields.

### 4.3. Scanning Electron Microscopy (SEM)

A scanning electron microscope TESCAN SOLARIS FE-SEM (Tescan, Czech Republic) was used to visualize surfaces (n = 6 in each group). All the samples were fixed on a conductive carbon tape. The 20 kV accelerating voltage in secondary electrons (SE) scanning mode was used. The conductive layer was not coated (native conditions). Twenty SEM fields 10 μm × 10 μm were analyzed for each sample.

### 4.4. Cytocompatibility Evaluation

#### 4.4.1. Materials

Samples of NiTi coated with TiO_2_ or TiO_x_N_y_ were sterilized for 30 min by incubation in 70 vol% aqueous ethanol, then washed three times with sterile phosphate-buffered saline (PBS). The bare NiTi samples were the control group. The EA.hy926 endothelial cells were kindly provided by Dr. C. J. Edgell (Carolina University, NC, USA). The EA.hy926 cells were grown in DMEM/F12 (Gibco, Carlsbad, CA, USA) medium supplemented with 10% fetal calf serum (FCS; Hyclone, Logan, UT, USA), 40 μg/mL gentamicin sulfate (Dalkhimpharm, Khabarovsk, Russia), and 2 mM L-glutamine (ICN, Costa Mesa, CA, USA) in a humid atmosphere with 5 vol% CO_2_ at 37 °C until a confluent monolayer was formed. The cells were cultured in flasks and removed with Trypsin-Versene (Biolot, Saint Petersburg, Russia) during passaging.

#### 4.4.2. Indirect Cytotoxicity Evaluation

The bare NiTi and 16 coated samples were incubated in 2.304 mL of the complete growth medium (DMEM/F12 supplemented with 10% FCS, 2 mM L-glutamine, and 40 μg/mL gentamicin) for 72 h at 37 °C, and had a surface area/volume ratio of 1.25 cm^−1^. Subsequently, the samples were removed from the medium, and the extracts were used to determine cytotoxicity against EA.hy926 cells. Specifically, the cells were seeded in a 96-well cell culture plate at 1 × 10^4^ cells per well and incubated for 24 h to allow attachment. The medium was then removed, and 100 μL of the extract was added, followed by culturing for another 24 h or 72 h. Cell viability was then determined using the 3-(4,5-dimethylthiazol-2-yl)-2,5-diphenyl-2H-tetrazolium bromide (MTT) assay (Sigma-Aldrich, Darmstadt, Germany) according to the manufacturer’s instructions. MTT (10 μL, 5 mg/mL) was added to each well, and incubation was continued for another 4 h. The formazan crystals formed after 4 h in each well were dissolved in dimethyl sulfoxide (150 μL, PanReac AppliChem, Darmstadt, Germany). The absorbance of the dissolved formazan crystals was measured at 492 nm using a Stat Fax-2100 multiwall plate reader (Awareness Technology, Inc., USA). Cell viability was calculated using the equation:Viability (%) = (OD_experimental group_/OD_control group_) × 100%(1)
where OD is the optical density of the samples, and the control group was cells cultivated in the DMEM/F12 medium without added extracts.

#### 4.4.3. Evaluation of Cell Adhesion

The EA.hy926 cells were exposed to the surfaces of coated NiTi samples (5 × 10^4^ cells in 20 μL, per sample), cultured for six days, and stained with phalloidin and 40,6-diamidino-2-phenylindole (DAPI, Abcam, Cambridge, UK) to expose the actin cytoskeleton. Phalloidin was conjugated to Alexa Fluor 488 (Thermo Fisher Scientific, Waltham, MA, USA) and incubated at a dilution of 1:200 (in PBS) for 1 h according to the manufacturer’s instructions. The surfaces of the samples were then imaged using an Axio Observer microscope (Zeiss, Oberkochen, Germany). Cell quantification was performed using at least five microscopic images of each surface, and the results were expressed as the number of adhered cells per 1 mm^2^.

#### 4.4.4. Cell Proliferation

Coated nitinol samples were placed into the wells of a 24-well plate (one sample per well) and exposed to cells (5 × 10^4^ cells in 20 μL, per sample), followed by cultivation for three and six days. The numbers of cell nuclei were determined by staining 40,6-diamidino-2-phenylindole (DAPI, Abcam, Cambridge, UK). The samples were examined using an Axio Observer microscope (Zeiss, Oberkochen, Germany). Cell nuclei were counted on abraded and polished surface of the samples after three and six days. Proliferation index (PI) of EA.hy926 cells on abraded and polished surface of coated NiTi samples was calculated using the equation:PI = number of cells on the 6th day/number of cells on the 3 day (2)

#### 4.4.5. Cell Viability Assay

TiO_2_- and TiO_x_N_y_-coated samples were placed into the wells of a 24-well plate (one sample per well) and exposed to cells (5 × 10^4^ cells in 20 μL, per sample), followed by cultivation for three and six days. The numbers of live and dead cells were determined by staining with fluorescent dyes, namely acridine orange (DIA M, Russia; 100 µg mL^−1^) and propidium iodide (Medigen, Novosibirsk, Russia; 100 µg/mL), which stain live and dead cells, and only dead cells, respectively. The samples were then incubated for 10 min at 37 °C and examined using an Axio Observer microscope (Zeiss, Oberkochen, Germany). For the assay, at least 500 cells were counted.

#### 4.4.6. NO Production Assay

NO production was assessed by measuring the levels of nitrite as a stable end product using the Griess reagent (Sigma-Aldrich, Taufkirchen, Germany) according to the manufacturer’s instructions. EA.hy926 cells were seeded on the surface of the sample in the wells of a 24-well plate (5 × 10^4^ cells per sample). After 24 h of cultivation, 50 μL of the supernatant was harvested and 50 μL of the Griess reagent were added to a 96-well plate. Absorbance at 492 nm was measured using a microplate reader (Stat FAX-2100, Awareness Technology Inc., Palm City, FL, USA), and nitrite concentrations were estimated using a standard calibration curve.

### 4.5. Statistical Analysis

Statistical analysis was performed using STATISTICA 10.0 software (StatSoft, Tulsa, OK, USA). Since the distribution of quantitative surface characteristics in most groups was not normal, non-parametric statistics were used, and data are reported as medians (Me) and interquartile ranges (25–75%) (IQRs). The data of the cell culture tests are presented as a mean (M) ± standard deviation (SD). The Mann–Whitney (M-W) U-test was used to compare two groups. The significance level was set to *p* < 0.05 (Appendix A).

## 5. Conclusions

The degree of smoothing of the substrate surface during pretreatment affects the relief of magnetron-sputtered TiO_2_ and TiO_x_N_y_ thin films, and this is expressed not only in a decrease in the height of nanopillars and RMSRs, but also in a change in the shape and density of nanopillars.Optimal compatibility with endothelial cells is ensured by the surface of N-TiO_2_ film sputtered on the abraded surface at N_2_:O_2_ = 1:1 and U_b_ = 0 V. This surface has a uniform density (~ 0.5 per 1 µm^2^) of nanopillars. The height of the peaks is from 618 nm to 1400 nm, with a mean summit curvature of 0.63.The most unfavorable characteristics for cell/surface interface are high S_ku_ with low S_ds_ and S_sc_.

## Figures and Tables

**Figure 1 ijms-24-06699-f001:**
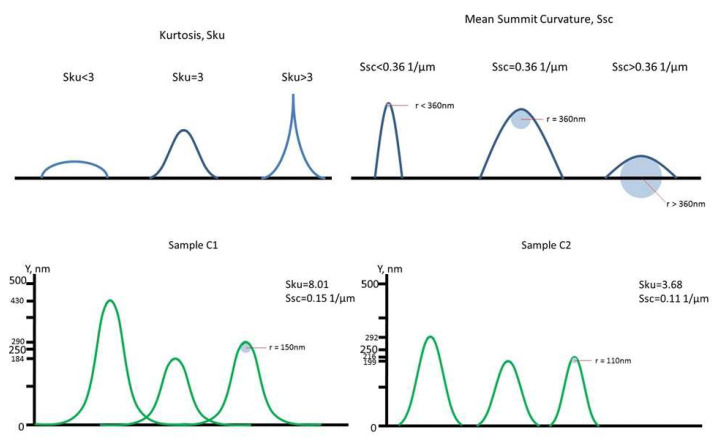
Examples of roughness measurement interpretation.

**Figure 2 ijms-24-06699-f002:**
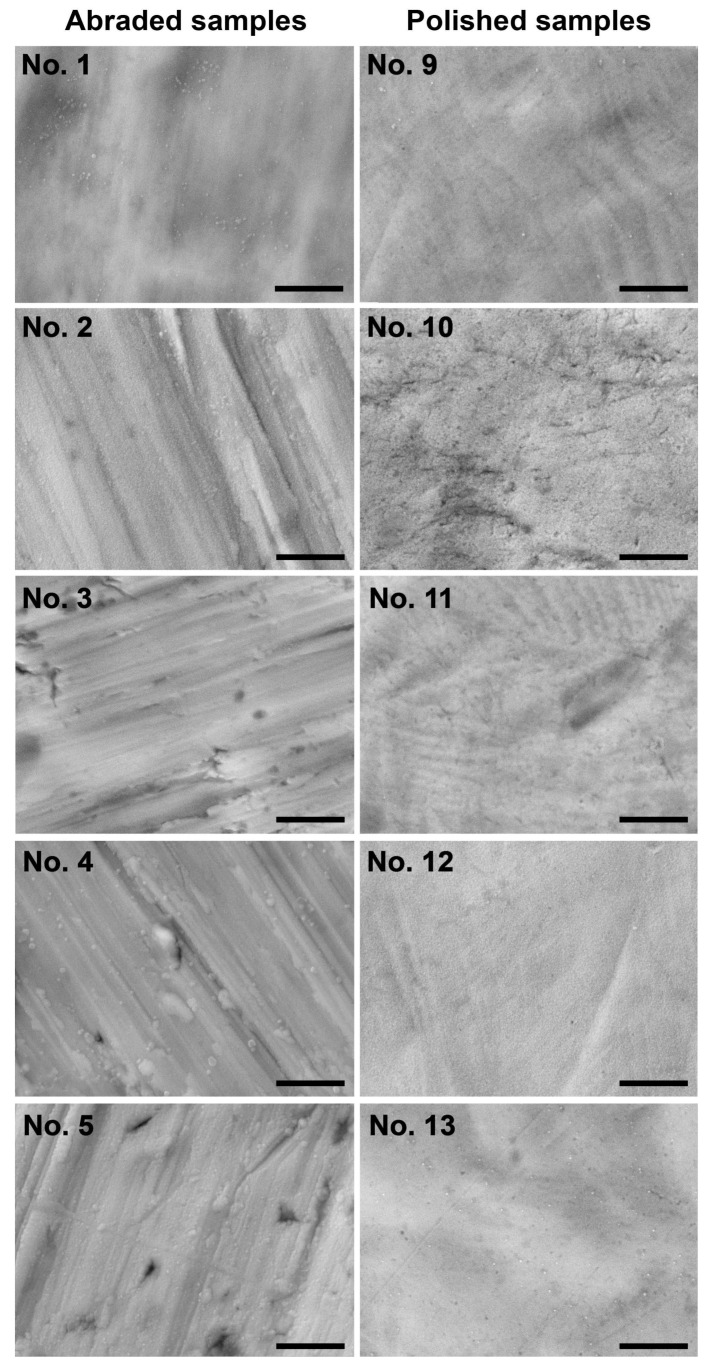
SEM images of the abraded (left column) or polished (right column) sample surfaces. The numbering of the samples is identical to Table 1. Scale bars are 2 μm.

**Figure 3 ijms-24-06699-f003:**
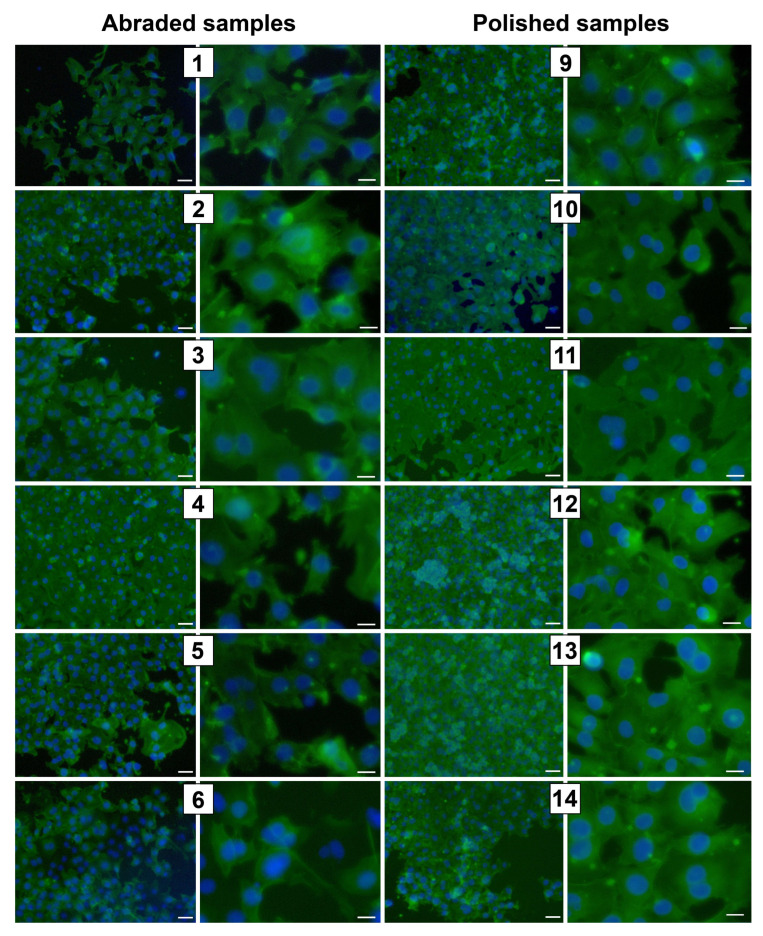
Fluorescence microscopy images of EA.hy926 cells on abraded or polished TiO_x_- and TiO_x_N_y_-coated samples after six-day culturing. Staining was performed with phalloidin (actin filaments, green) and DAPI (nuclei, blue). Scale bar: 50 μm (left column) and scale bar: 20 μm (right column).

**Figure 4 ijms-24-06699-f004:**
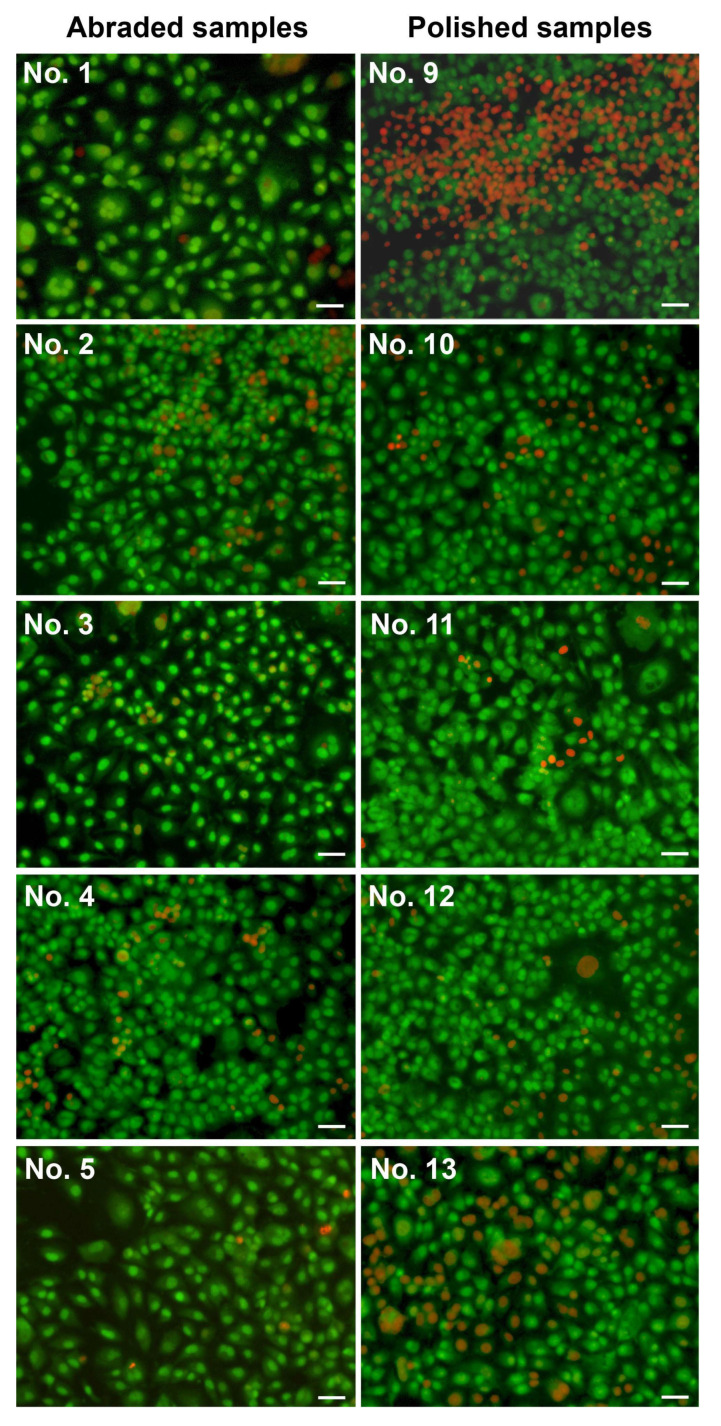
Fluorescence microscopy images of EA.hy926 cells on abraded or polished TiOx and TiOxNy-coated samples after six-day culturing. Staining was performed with acridine orange (green, live cells) and propidium iodide (red, dead cells). Scale bar: 50 μm.

**Figure 5 ijms-24-06699-f005:**
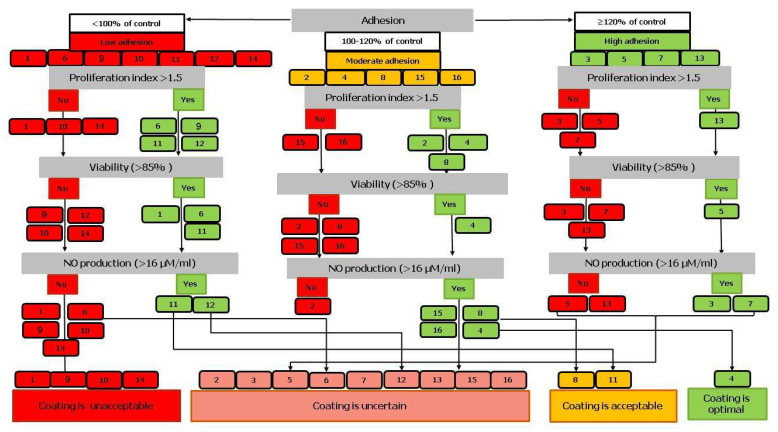
Decision tree algorithm for the cytocompatibility of abraded and polished TiO_2_ and TiO_x_N_y_ samples.

**Table 1 ijms-24-06699-t001:** Details of the employed deposition modes.

Group No.		Partial Gas Pressure, Pa, N_2_/O_2_	Negative Bias Voltage	Films	N_2_/O_2_ Ratio
Abraded	Polished				
1	9	0/0.130	U = −100 V	TiO_2_	-
2	10	0/0.130	U = 0	TiO_2_	-
3	11	0.065/0.065	U = −100 V	Ti-O-N	1/1
4	12	0.065/0.065	U = 0	Ti-O-N	1/1
5	13	0.087/0.046	U = −100 V	Ti-O-N	2/1
6	14	0.087/0.046	U = 0	Ti-O-N	2/1
7	15	0.102/0.033	U = −100 V	Ti-O-N	3/1
8	16	0.102/0.033	U = 0	Ti-O-N	3/1
C1 (NiTi)	C2 (NiTi)	-	-	-	-

**Table 2 ijms-24-06699-t002:** Surface roughness characteristics *.

No.	S_q_, nm	S_p_, nm	S_z_, nm
C1	65.28 (47.40; 74.18)	290.53 (184.33; 430.70)	795.94 (418.07; 977.06)
C2	41.95 (39.79; 45.61)	216.92 (199.26; 292.09)	391.49 (347.36; 521.28)
1	83.00 (53.39; 95.81)	301.12 (261.76; 387.79)	676.24 (415.84; 708.05)
2	81.55 (68.78; 91.11)	400.34 (302.28; 572.13)	758.53 (567.05; 1079.99)
3	104.56 (99.38; 119.01)	357.15 (327.13; 401.46)	969.62 (778.01; 1087.29)
4	127.42 (110.28; 147.00)	711.69 (618.62; 1399.00)	1473.64 (1150.95; 2203.00)
5	111.59 (95.90; 140.80)	588.99 (479.60; 799.83)	1330.24 (1125.72; 1352.33)
6	97.73 (90.55; 101.67)	360.63 (338.30; 445.78)	858.68 (785.93; 1047.61)
7	69.60 (58.42; 72.73)	348.65 (324.20; 369.25)	749.23 (675.25; 884.06)
8	70.66 (67.36; 83.21)	360.95 (344.48; 392.69)	796.64 (653.91; 1125.42)
9	59.35 (54.32; 64.96)	226.25 (218.58; 239.61)	421.50 (396.91; 446.40)
10	61.73 (60.68; 71.66)	300.27 (226.83; 355.49)	525.52 (417.04; 600.62)
11	62.46 (59.81; 64.97)	284.18 (273.18; 292.35)	506.40 (444.63; 509.07)
12	51.84 (48.18; 58.48)	277.71 (196.31; 465.75)	504.69 (362.60; 686.68)
13	66.55 (66.17; 71.30)	339.89 (233.93; 359.63)	613.92 (421.85; 637.11)
14	65.81 (62.44; 83.06)	300.02 (259.29; 313.50)	836.09 (810.07; 1072.13)
15	62.44 (59.92; 68.92)	313.50 (286.37; 319.30)	515.08 (487.47; 810.07)
16	55.82 (55.20; 62.83)	263.64 (263.64; 296.20)	730.73 (623.55; 730.73)

* S_q_—root mean square roughness of the surface; S_p_—maximum peak height, nm; S_z_—maximum height of the surface, nm.

**Table 3 ijms-24-06699-t003:** Characteristics of the surface nanopeaks’ shape.

No.	S_ku_, Kurtosis of Height Distribution	S_sc_, Mean Summit Curvature, 1/μm	S_ds_, Density of Summits of the Surface, 1/μm^2^
C1	8.01 (3.51; 11.10)	0.15 (0.14; 0.26)	0.48 (0.38; 0.60)
C2	3.68 (2.81; 4.06)	0.11 (0.11; 0.12)	0.29 (0.26; 0.30)
1	2.93 (2.53; 3.42)	0.15 (0.10; 0.17)	0.16 (0.11; 0.16)
2	4.34 (2.94; 5.01)	0.45 (0.36; 0.47)	0.58 (0.55; 0.71)
3	3.57 (2.93; 4.05)	0.45 (0.08; 0.66)	0.55 (0.10; 0.66)
4	4.44 (3.85; 5.00)	0.63 (0.61; 0.67)	0.49 (0.47; 0.53)
5	3.90 (3.30; 4.43)	0.49 (0.46; 0.52)	0.52 (0.50; 0.55)
6	3.72 (3.14; 3.78)	0.30 (0.12; 0.31)	0.17 (0.12; 0.37)
7	3.83 (3.70; 4.03)	0.37 (0.32; 0.37)	0.38 (0.36; 0.42)
8	5.04 (4.58; 5.80)	0.35 (0.23; 0.43)	0.36 (0.27; 0.51)
9	3.19 (2.94; 3.49)	0.22 (0.11; 0.29)	0.38 (0.23; 0.51)
10	4.16 (3.45; 5.22)	0.04 (0.04; 0.05)	0.08 (0.06; 0.10)
11	3.35 (2.97; 3.83)	0.23 (0.17; 0.32)	0.25 (0.18; 0.27)
12	3.44 (2.80; 3.74)	0.16 (0.11; 0.29)	0.43 (0.27; 0.52)
13	3.45 (2.95; 3.90)	0.40 (0.28; 0.41)	0.39 (0.31; 0.42)
14	6.22 (5.29; 6.39)	0.56 (0.54; 0.57)	0.74 (0.70; 0.75)
15	3.22 (3.21; 4.32)	0.57 (0.28; 0.58)	0.39 (0.21; 0.72)
16	4.42 (3.52; 4.42)	0.54 (0.54; 0.54)	0.93 (0.80; 0.94)

**Table 4 ijms-24-06699-t004:** The viability of endothelial cells exposed to sample extracts, %.

Abraded Samples			Polished Samples		
No.	24 h	72 h	No.	24 h	72 h
1	82.19 ± 7.52 *#	86.92 ± 12.71 #	9	110.10 ± 6.24	93.52 ± 6.16
2	112.85 ± 6.78 *#	103.34 ± 16.72	10	100.49 ± 2.01 #	100.57 ± 1.09
3	95.71 ± 6.60 *	89.56 ± 9.97 #	11	107.19 ± 1.60	103.82 ± 6.13
4	106.78 ± 8.73	93.19 ± 10.60 #	12	112.43 ± 6.87	102.10 ± 7.94
5	100.55 ± 15.07	89.76 ± 11.03 *#	13	105.33 ± 4.60	109.23 ± 3.37 #
6	98.35 ± 5.38	110.97 ± 12.41 *	14	101.19 ± 5.86	85.18 ± 3.39 #
7	100.83 ± 7.40	122.66 ± 7.44 *	15	104.86 ± 5.67	99.33 ± 0.78
8	94.81 ± 6.55 *	109.20 ± 15.22	16	107.02 ± 6.41	104.11 ± 1.22 #
NiTi (C1)	101.15 ± 8.05 *	125.70 ± 26.75	NiTi (C2)	115.05 ± 12.18	99.33 ± 3.25

Data (n = 6 in each group) are presented as M ± SD. * indicates significant differences between the values of abraded and polished surfaces (*p* < 0.05), # indicates significant differences between the values coated and bare NiTi samples (*p* < 0.05). These notes are identical for Appendix A.

**Table 5 ijms-24-06699-t005:** Number of adhered EA.hy926 cells on the sample surfaces, cells/mm^2^.

Abraded Samples			Polished Samples		
No.	3 days	6 days	No.	3 days	6 days
1	456.82 ± 68.63 *#	441.97 ± 77.63 *	9	714.02 ± 128.71	1441.04 ± 594.89 #
2	606.13 ± 53.75	967.97 ± 248.94 #	10	727.02 ± 166.23	889.14 ± 208.54 #
3	705.85 ± 35.76 #	622.28 ± 277.54	11	621.63 ± 14.55 #	847.35 ± 65.66 #
4	630.45 ± 106.97 *	1270.89 ± 364.02 #	12	202.55 ± 65.77	1393.87 ± 373.21 #
5	768.80 ± 123.36 #	645.68 ± 145.51 *#	13	1046.43 ± 217.34	1653.48 ± 484.69 #
6	38.07 ± 14.28 *#	509.19 ± 120.93 #	14	691.36 ± 123.53	747.91 ± 182.55 #
7	1075.21 ± 152.47 #	1192.76 ± 142.20 #	15	888.12 ± 279.09	876.04 ± 275.26 #
8	645.13 ± 178.55	950.97 ± 189.85 #	16	831.75 ± 218.04	854.22 ± 191.81 #
NiTi (C1)	542.25 ± 121.62	390.67 ± 72.69 *	NiTi (C2)	777.56 ± 193.56	81.85 ± 45.93

Data (n = 6 in each group) are presented as M ± SD. * indicates significant differences between the values of abraded and polished surfaces (*p* < 0.05), # indicates significant differences between the values coated and bare NiTi samples (*p* < 0.05).

**Table 6 ijms-24-06699-t006:** Proliferation index of EA.hy926 cells on abraded and polished surface of samples.

No.	Abraded Samples	No.	Polished Samples
1	0.96 ± 0.22 *	9	2.31 ± 1.10 #
2	1.57 ± 0.48 #	10	1.29 ± 0.45 #
3	0.90 ± 0.43 *	11	1.51 ± 0.23 #
4	1.95 ± 0.50 *#	12	6.27 ± 2.00 #
5	0.87 ± 0.1 *	13	2.56 ± 2.25 #
6	17.40 ± 11.60 *#	14	1.05 ± 0.31 #
7	1.14 ± 0.26 #	15	1.08 ± 0.66 #
8	1.60 ± 0.62 *#	16	1.05 ± 0.23 #
NiTi (C1)	0.62 ± 0.15	NiTi (C2)	0.12 ± 0.05 #
EA.hy926	3.86 ± 0.104	EA.hy926	3.86 ± 0.108

Data (n = 6 in each group) are presented as M ± SD. * indicates significant differences between the values of abraded and polished surfaces (*p* < 0.05), # indicates significant differences between the values coated and bare NiTi samples (*p* < 0.05).

**Table 7 ijms-24-06699-t007:** Viability of the EA.hy926 cells on abraded and polished surfaces samples, %.

Abraded Samples			Polished Samples		
No.	3 days	6 days	No.	3 days	6 days
1	83.99 ± 17.98 *	87.36 ± 7.54 *#	9	16.92 ± 16.87 #	64.12 ± 17.60 #
2	40.20 ± 15.11 *#	75.81 ± 9.20	10	78.52 ± 8.02	69.74 ± 15.29
3	34.32 ± 10.40 *#	82.97 ± 5.82 #	11	82.23 ± 9.35 #	87.27 ± 5.10
4	25.29 ± 7.16 *#	88.74 ± 3.28 *#	12	99.80 ± 0,09 #	70.42 ± 17.32
5	51.73 ± 10.93 *#	94.35 ± 2.27 *#	13	8.12 ± 18.13 #	67.73 ± 15.46
6	7.93 ± 2.15 #	87.46 ± 2.73 *#	14	0.59 ± 0.91 #	5.21 ± 7.08 #
7	28.17 ± 6.53 *#	84.05 ± 4.89 *#	15	44.87 ± 12.07 #	60.06 ± 20.07 #
8	11.40 ± 5.54 *#	83.05 ± 6.25 *#	16	81.63 ± 12.82	71.60 ± 9.64
NiTi (C1)	77.88 ± 11.06 *	58.22 ± 18.94 *	NiTi (C2)	65.80 ± 12.45	83.92 ± 14.21

Data (n = 6 in each group) are presented as M ± SD. * indicates significant differences between the values of abraded and polished surfaces at 3 days and 6 day (*p* < 0.05), # indicates significant differences between the values coated and bare NiTi samples at 3 days and 6 day (*p* < 0.05).

**Table 8 ijms-24-06699-t008:** NO production of EA.hy926 cells on abraded and polished surfaces samples, μM/mL.

No.	Abraded Samples	No.	Polished Samples
1	14.55 ± 3.13	9	13.43 ± 2.58
2	13.51 ± 2.55	10	15.01 ± 4.05
3	17.68 ± 0.48	11	18.25 ± 1.66
4	22.05 ± 5.36	12	28.97 ± 8.36 #
5	9.76 ± 3.19 #	13	8.15 ± 1.22 #
6	12.47 ± 2.44 #	14	11.32 ± 3.52 #
7	56.22 ± 31.38 #	15	49.17 ± 15.19 #
8	52.16 ± 13.70 #	16	48.54 ± 18.48 #
NiTi	16.06 ± 2.43	NiTi	15.13 ± 5.11
EA.hy926	17.23 ± 3.34	EA.hy926	17.23 ± 3.34

Data (n = 6 in each group) are presented as M ± SD. “#” indicates significant differences between the values coated and bare NiTi samples (*p* < 0.05).

**Table 9 ijms-24-06699-t009:** Ranking the assessment of the biological test results.

Criterion	Parameter		
	Unacceptable Level	Moderate Level	Optimal Level
Cell adhesion, % (100% is adhesion on bare NiTi surface on the third day)	<100% of bare NiTi control level low adhesion	100–120% of bare NiTi control level moderate adhesion	>120% of bare NiTi control level high adhesion
Proliferation index	<1.5 low proliferation	-	>1.5 high proliferation
Cell viability, % (relative content of living cells on the sixth day)	<85% non-cytocompatible surface	-	≥85% cytocompatible surface
NO production, μM/mL (16 μM/mL—NO production level on the C1 surface)	<16 μM/mL low functional activity	-	>16 mM/mL high functional activity

## Data Availability

Not applicable.

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
