# Peer review of "Effect of the Nanorough Surface of TiO2 Thin Films on the Compatibility with Endothelial Cells"

_ijms, 2023, doi:10.3390/ijms24076699_

Round 1
Reviewer 1 Report
Overall comments:
The subject matter of this paper deals with the investigation of the influence of the surface nanostructure of TiO2 films on the cytocompatibility of endothelial cells. For this purpose, the authors prepared 16 groups of samples with varying surface topography, including eight hand-abraded and eight machine-polished. They showed that the excessive sharpness of surface nanostructures during magnetron sputtering of TiO2 and N-TiO2 films, especially at high density, may damage the cell membrane and prevent efficient adhesion and proliferation of endothelial cells.
The manuscript itself is considered to be theoretically and structurally reasonable. However, there are some specific concerns that should be clearly addressed. If such issues were all cleared by the authors, this paper seems to be qualified to secure its publication.
Specific concerns:
1. The manuscript contains numerous typos, including issues with spacing and upper/lower case. It is recommended to meticulously review the entire manuscript to correct these errors.
2. It is suggested to split Figure 2 into two separate figures based on the results of the live/dead assay and phalloidin staining, respectively. Furthermore, the term "fhalloidin" should be corrected to "phalloidin."
3. To better understand the morphology and F-actin structures that are crucial for cell growth and fate, it is recommended to include higher magnification images of cells during early culture periods. The cells in Figure 2 are too densely packed, making it difficult to discern these important details.
Author Response
- The manuscript contains numerous typos, including issues with spacing and upper/lower case. It is recommended to meticulously review the entire manuscript to correct these errors.
Thanks for your comments. We have made corresponding revision according to your suggestion and corrected the errors.
- It is suggested to split Figure 2 into two separate figures based on the results of the live/dead assay and phalloidin staining, respectively. Furthermore, the term "fhalloidin" should be corrected to "phalloidin."
Following the Reviewer’s recommendations, Figure 2 was divided into two figures based on the results of phalloidin staining (Figure 3) and live/dead analysis (Figure 4). In Fig.3 we have included the higher magnification images.
- To better understand the morphology and F-actin structures that are crucial for cell growth and fate, it is recommended to include higher magnification images of cells during early culture periods. The cells in Figure 2 are too densely packed, making it difficult to discern these important details.
The reviewer is absolutely right that F-actin structures are crucial for cell growth and fate, so it is important to study the cytoskeleton at the early stages of adhesions (e.g., within 4-24 hours). Studying the early stages of the adhesions is the subject of further research, but it is beyond the scope of the present work. To shed some light on this point within the present work, we included high magnification images after 6 days of culture in Figure 2.
Reviewer 2 Report
In the manuscript entitled “Effect of the Nanorough Surface of TiO2 Thin Films on the Compatibility with Endothelial Cells”, Mayorov and co-workers used magnetron sputtering to manipulate the surface morphologies to study the influence on endothelial cells. Several different surfaces with various morphologies were fabricated, which served as interesting platform to elucidate the surface-cell interactions. Nevertheless, there are some concerns on the experimental design. I would suggest the revision of the manuscript before further consideration for International Journal of Molecular Sciences.
1. Magnetron sputtering represents an interesting strategy to tune the surface property and serves as an alternative for 3D printing and lithography methods. The authors should further discuss the advantages of this method in their introduction and discussion section. How to improve the reproducibility of this method?
2. While AFM images in SI provided useful information, it is recommended to use SEM to conform the surface morphology and offer more “visualized” images for the surface.
3. For the proliferation experiment and the NO production experiment, the authors should include proper control groups to check the background signal and clarify the impact of surface morphology. C1 and C2 are not enough.
Some minor suggestions:
1. Some viability data in Table 5 exceeds 100%. Please explain.
2. The presentation of results with tables is not intuitive. It is recommended to use figures (either columns or lines) to better present the data.
Author Response
We thank the reviewers for acknowledging our work and for giving us insightful comments. We have responded to the comments below in a point-by-point manner and made corresponding revisions in the text.
- Magnetron sputtering represents an interesting strategy to tune the surface property and serves as an alternative for 3D printing and lithography methods. The authors should further discuss the advantages of this method in their introduction and discussion section. How to improve the reproducibility of this method?
It was done. The revised text is highlighted in yellow in the revised manuscript.
- While AFM images in SI provided useful information, it is recommended to use SEM to conform the surface morphology and offer more “visualized” images for the surface.
Following the reviewer’s recommendation, we added the SEM characterization of the thin film surfaces (Fig.2 and the text).
- For the proliferation experiment and the NO production experiment, the authors should include proper control groups to check the background signal and clarify the impact of surface morphology. C1 and C2 are not enough.
Following the reviewer’s recommendation, the endothelial cell proliferation index on cultured plastic was added in the revised manuscript as an appropriate control group for the proliferation experiment. Also, the NO production of endothelial cells cultured on plastic was added as an appropriate control group for the NO production assay (Tables 7, 9 and Figures S12, 14).
Some minor suggestions:
- Some viability data in Table 5 exceeds 100%. Please explain.
The MTT test allows one to assess not only the viability, but also the proliferative activity of cells. Cell viability measurement by MTT is based on the ability of metabolically active cells to reduce the tetrazolium salt into purple formazan crystals, which can be determined by spectrophotometry. The results of the MTT test reflect the activity of mitochondria and are proportional to the number of cells. [https://doi.org/10.3390/ijms222312827]. Therefore, an increase in the viability of endothelial cells with the addition of extracts of samples (Table 5) above 100% indicates not only the absence of cytotoxicity of the studied samples, but also an increase in the proliferative activity of cells relative to the control (endothelial cells without the addition of extracts).
- The presentation of results with tables is not intuitive. It is recommended to use figures (either columns or lines) to better present the data.
We are sure that the tables can present the most accurate information about the cell tests’ data. Following the reviewer's recommendation, we added the Figures in the Supplementary (Figures S10-S14) of the revised manuscript, which illustrate the data of the tables.